# The Importance of Safety and Security Measures at Sharm El Sheikh Airport and Their Impact on Travel Decisions after Restarting Aviation during the COVID-19 Outbreak

**Thowayeb H. Hassan [1,2,\*] and Amany E. Salem [1,2]**

[1] College of Arts, King Faisal University, Al-Ahsa P.O. Box 31982, Saudi Arabia; asalem@kfu.edu.sa or amany.ibrahim@fth.helwan.edu.eg
[2] Faculty of Tourism and Hotel Management, Helwan University, Cairo P.O. Box 12612, Egypt
[\*] Correspondence: thassan@kfu.edu.sa or thowayeb.hassan@fth.helwan.edu.eg; Tel.: +966-54-029-4550

**Abstract:** Travel decisions during the COVID-19 pandemic might be substantially influenced by destination-based attributes, in particular, health safety measures at airports. In the current study, we aimed to assess the effects of the perceived importance of safety measures at the Sharm El Sheikh airport on the intention of international passengers to revisit the destination, which might reflect their behavioral control for traveling to other tourism destinations. A total of 954 international travelers were asked to fill out a survey to reveal their travel risk perceptions, the importance of airport safety measures, and their future intentions to revisit the destination, and the data were integrated in an SEM model. The results showed that passengers with low-risk perceptions and highly perceived importance of logistic and sanitization procedures, as well as traveler- and staff-related safety measures, were more likely to exhibit greater intentions to revisit the city and lower intentions to cancel or change future travel plans to other touristic regions. Health safety at airports should be stressed in future strategic plans by governmental authorities and stakeholder activities to mitigate the psychological barriers of tourists.

**Keywords:** travel behavior; intention to travel; COVID-19; safety measures; air transport; sustainable transport systems

## 1. Introduction

The novel coronavirus disease (COVID-19) has caused formidable and unprecedented challenges in multiple sectors worldwide since the first description of the outbreak as a pandemic on 12 March 2020 [1]. The aviation sector has been no exception due to the inherent vulnerability of the global air transport to disruptive events. Indeed, aircraft and airline passengers were first described as vectors of human infectious diseases in the late 1920s, when commercial flights between European countries and destinations in the Middle East, Africa, and India began to operate [2,3]. The sheer volume of traveler movement has supported the "mobility turn" theory, which provoked a plethora of studies to investigate the multidimensional aspects of aviation security [4,5]. Nevertheless, multiple dimensions of sanitary and safety regulations during disruptive infectious instances have been scarcely considered in the academic literature.

In the recent past, the commercial aviation sector has been influenced by four major outbreaks, including SARS, EBOLA, bird flu, and H1N5 influenza [6–8]. Actually, such outbreaks lasted for relatively short time periods, were restricted to distinct regions, and they exhibited low rates of symptomatic infections. Consequently, there were no widespread global travel restrictions, national lockdowns, border closures, or rigorous quarantine measures. Furthermore, passengers' perceptions and willingness to travel did not have a significant impact on the travel sector. Contrastingly, the COVID-19 outbreak has been associated with global flight restrictions, closed borders, and strict quarantine periods,

leading to a rapid decline in international and domestic tourism worldwide. Within the space of months, the global system has evidenced a significant transition from overtourism to non-tourism [9,10]. As a consequence, according to the International Air Transport Association [11], it is expected that the pre-pandemic levels of passenger demand will only be attained in 2023 at the earliest.

Therefore, it is necessary to investigate the self-perceptions of passengers regarding their future intentions to revisit a destination. Regardless of the scientific rationale, the perceived efficacy of safety measures for air travel should inevitably influence travelers' decision making. The use of face masks, thermo screening, sanitization procedures, pre-flight testing, social distancing, and air filtration systems reflect efforts that are aimed at reducing the likelihood of exposure to aninfection. Such measures seem to be important in making future decisions, in particular, with the lack of unanimity in regard to definitive risk areas, methods of virus transmission, and the ideal thresholds of restricted travel and quarantine measures. After re-opening the borders to international tourism, the subjective assessment of tourists' behavioral intentions to revisit the destination is crucial, such that airlines and tourism companies can focus on specific domains that ensure reducing infection transmission and, at the same time, satisfy tourists' needs for better attachment to the destination. In Egypt, after flight suspension in March 2020, hotels in Sharm el-Sheikh resumed their activities on 15 May 2020. Domestic flights were only allowed until mid-July, when regional and international flights returned [12]. This was associated with strict health and safety requirements at the airport. In this context, the aim of the present study was to assess the perceived importance of safety and infection control measures at airports from the perspectives of international tourists at the Sharm el-Sheikh airport. The impact of adopting such measures on the behavioral intentions to revisit the destination and the intentions to cancel or change travel plans to other regions was also evaluated.

## 2. Literature Review

### 2.1. The Perceived Risks of Infectious Diseases and Risk Reduction Strategies

Tourists' decision making to revisit a destination has been cited as a major determinant of the sustainability of tourism and travel establishments [13]. Future behavioral intention is referred to as the degree to which an individual tends to perform or not perform a distinct action [14]. In the tourism sector, the intention of tourists can be subjectively evaluated through a survey-based approach to collect personal data for assisting the likelihood of returning to COVID-19-affected destinations. After re-opening the Egyptian borders to international tourism, tourist's perceptions of risk play an important role in future decisions. Tourists perceive risk differently based on their personal knowledge, their degree of exposure to risks, and their risk acceptance levels [15] and their perceived risks may also differ significantly according to their nationality, religious backgrounds, cultural orientation, and psychographic characteristics [16–18]. When their risk tolerance levels reach a certain threshold, tourists either abandon the trip or engage in risk reduction strategies to alleviate the effect of the perceived risk.

Generally speaking, the adaptive strategies of tourists fall under two major categories, i.e., modification of the consumption behavior and information search [15]. While the former deals with the adjustment of behavioral strategies to avoid or reduce the impact of a risk, the latter is related to gathering information on the best ways to reduce the risk. Tourists, as expected, integrate various types of information about the recent situation of COVID-19 spread and the relevant preventive measures at a destination. This is in line with the information integration theory [19], which indicates that individuals are more likely to search and integrate information from a number of sources in order to finally attain a judgement about a situation, a person, or an object. Accordingly, more favorable judgements are based on highly valued information. However, uncertainty avoidance theory states that individuals with negative or relatively uncomfortable attitudes towards an unknown or an ambiguous situation attempt to reduce uncertainty via looking for information from trusted sources [20]. Given that tourists might be exposed to infection at

a destination with high risk of transmission, tourists seek information about the measures adapted to ensure travelers' safety.

In the literature, health risks have been frequently cited as a significant determinant of travel decision making. For instance, backpackers arriving at the Kotoka International Airport in Ghana expressed significant interest in information for avoiding poor sanitary conditions and exposure to tropical infections at their travel decisions [15]. Similarly, infectious diseases were independent predictors of the intention to change travel plans as revealed in an early study involving passengers departing from the Hong Kong Airport [21]. Furthermore, infection control strategies, such as the requirement to wear face masks and health screening of tourists coming from endemic areas, were highly acknowledged by international tourists at the Bangkok International Airport during the SARS and bird flu outbreaks [22]. Consequently, the decision to visit and revisit a destination is mostly affected by perceived risk of infection based on an interaction among the acquired information, tourist attributes, and the characteristics of the destination [23].

Considering the context of the recent COVID-19 pandemic, it has been shown that the adoption of international travel restrictions is more effective for reducing the daily reported incoming cases rather than fully reopening and the implementation of voluntary quarantine measures (even at a rate of 95%) [24]. However, an early systematic review showed that international travel restrictions delayed the spread of human influenza epidemics by two months, and reduced the incidence of the disease by 3% [25]. Therefore, the impact of travel restrictions was only evident on delaying disease spread rather than preventing it. Alternatively, partial reopening is possible; however, it should be accompanied by strict preventive measures. Since recently-infected individuals are usually asymptomatic, strict quarantine conditions are required at their destination [26]. Therefore, starting from the first point of arrival, namely the airports, implementation of personal protective interventions seems to be effective for risk reduction.

### 2.2. Safety at Airports and Its Efficacy to Reduce the Rate of Infection

Basically, according to the *Oxford Learner's Dictionary* [27], the concept of safety is defined as the state of being protected from a distinct danger or harm. Within the context of airports and tourism, on the one hand, safety includes the protection of passengers physically. On the other hand, the image of the environment of a given destination is also protected [28]. Therefore, multiple strategies have been introduced to reduce the incidence of infectious diseases, their rates of transmission, and disease-specific morbidity and mortality due to international travel. For example, the application of a quarantine after travel is one of the oldest measures known. Quarantine is defined as the restriction of movement of apparently healthy individuals who have had exposed to a contagious disease. It requires integrated coordination of multiple sectors, the establishment of reliable communication channels, and the implementation of new legislative actions by the authorities [29]. A recent Cochrane systematic review indicated that the basic reproduction number of COVID-19 were reduced by 37–88% after implementing effective quarantine measures [30]. Furthermore, a combined model of safety measures, including quarantine, social distancing, and school closures has led to a significant reduction in COVID-19 incidence, transmission, and disease-related deaths as compared with the same measures without quarantine [30].

Screening procedures have also proven to be effective for limiting trans-border transmission of infectious diseases and onboard transmission. While COVID-19 is predominantly transmitted in the symptomatic phase of the infection [31,32], transmission via contacting asymptomatic individuals or presymptomatic patients may be possible [33]. As such, temperature screening procedures have been recommended by IATA [34]. Despite the high cost of installing thermoscanners, it has been anticipated that as high as 45% of travelers might be detected [35]. However, since there is a considerable proportion of infected individuals who do not develop fever, particularly among the young populations [36], the efficacy of these measurements might be relatively inadequate in airports [37].

To further control the spread of COVID-19, many countries have required that passengers possess a negative PCR test result, where the test should have been performed within a specific time period before arrival to the destination. Other countries have required PCT testing upon arrival regardless of prior test results at the origin [38]. Some simple measures have been frequently reported in different airports, such as allowing the entry of passengers only to the airport, mandating personal protective equipment for the working staff, and regular disinfecting of surfaces [39]. Wearing face masks, physical separation by leaving middle seats free, and air filtration in planes are also common measures. Indeed, air filtration is a critical intervention given that air travelers spend prolonged periods in enclosed spaces, and thus the risk of spread of infection is theoretically substantial. This has been confirmed by recent epidemiological investigations. For example, in January 2020, Chen, et al. [40] showed that 16 patients (out of 335 passengers) were diagnosed with COVID-19 after exposure to virus particles during a flight from Singapore to Hangzhou International Airport in China. Similar observations were reported in other flights between Sydney and Perth [41] and between London and Hanoi [42]. Consequently, the transmission of COVID-19 infection can be inhibited by employing strict preflight and onboard measures.

## 3. Materials and Methods

### 3.1. Research Context

Sharm El Sheikh is a major tourism city and a popular Red Sea resort in South Sinai Governorate, Egypt. The city has gained significant interest due to its strategic location at the narrow entry point to the Gulf of Aqaba. In addition, it has become an important destination in the global tourism sector given the unique biodiversity of the marine life of the Red Sea. Since the resumption of international air traffic to Sharm El Sheikh, more than one million tourists have arrived to the local airport [43,44]. All passengers on local and domestic flights are asked to provide a PCR report which proves that the passenger has tested negative for COVID-19 within 72 h before boarding. The Airports Council International (ACI) has recently granted the Health Accreditation Seal for Safe Travel to Sharm El-Sheikh International Airport, which reflects the dedication and efforts of the national aviation sector for ensuring the application of strict safety measures to combat the global pandemic [45]. Given such unique touristic aspects and proven safety measures, we selected Sharm El Sheikh as the destination of choice and the national airport as the study setting.

### 3.2. Measures and Data Collection

In this study, we employed a quantitative research design, where a self-administered, structured questionnaire was submitted to international travelers at Sharm El Sheikh Airport during the period between 1 November 2020 and 31 January 2021. The survey was uploaded on a specifically designated platform (Google Forms), and the respondents were invited to fill out the online questionnaire via travel agencies which prepared the programs for the tourists. International travelers who traveled to Sharm El Sheikh for leisure or work purposes were eligible. The instrument was comprised of five major domains (31 items), which included tourists' travel risk perception during the pandemic, perceived importance of safety measures at the airport, willingness to change or cancel travel plans, and personal intention to travel in the future. Items for all the aforementioned domains were ranked on a five-point Likert scale, ranging from 1 (strongly disagree) to 5 (strongly agree). The purpose of respondents' visits to Sharm El Sheikh as well as their demographic characteristics were additionally collected, including gender, age, level of education, and the frequency of travel before the pandemic. The included items were adapted from previous studies; the measurement of travel risk perception was based on six items [46,47], i.e., perceived importance of safety measures at the airport on 16 items, willingness to change or cancel travel plans on six items [48], and personal intention to travel in the future on three items [49].

### 3.3. Model Analysis

The statistical analysis was conducted using the Statistical Package for Social Sciences version 26.0 (SPSS Inc., Chicago, IL, USA) and AMOS v26. Demographic characteristics as well as the purpose of visiting the destination were expressed as frequencies and percentages. Likert responses of different items (from 1 to 5) were presented as means ± standard deviations (SDs). Dimension reduction, namely exploratory factor analysis (EFA), was performed on the "perceived importance of safety measures" domain to derive valid constructs from the included items ($n$ = 16). More specifically, a principal component analysis (PCA) technique was carried out to load the variables of interest into a smaller set of composite components with a shared variance [50]. The analysis was performed using an eigenvalue of 1 to improve the strength of the obtained factors [51]. In addition, the minimum factor loading was set at 0.3, and the rotation method was the Promax method with Kaiser normalization. As demonstrated in Table 1, a three-factor solution was extracted. The results of the PCA showed that the ratios of the unique variance to the shared variance (communality) were >0.2 for all items, and the four components explained 75.28% of the total variance (Figure 1). The Kaiser–Meyer–Olkin (KMO) measure was 0.892 and the Bartlett's test of sphericity was significant ($\chi^2$ = 15,441.630, $p$ < 0.0001), indicating a significant sampling adequacy. Ultimately, the items were loaded according to the following components: sanitization and logistics operations (7 items), staff- and traveler-related preventive measures (6 items), and innovative preventive measures (3 items). These components were further incorporated in the statistical analysis.

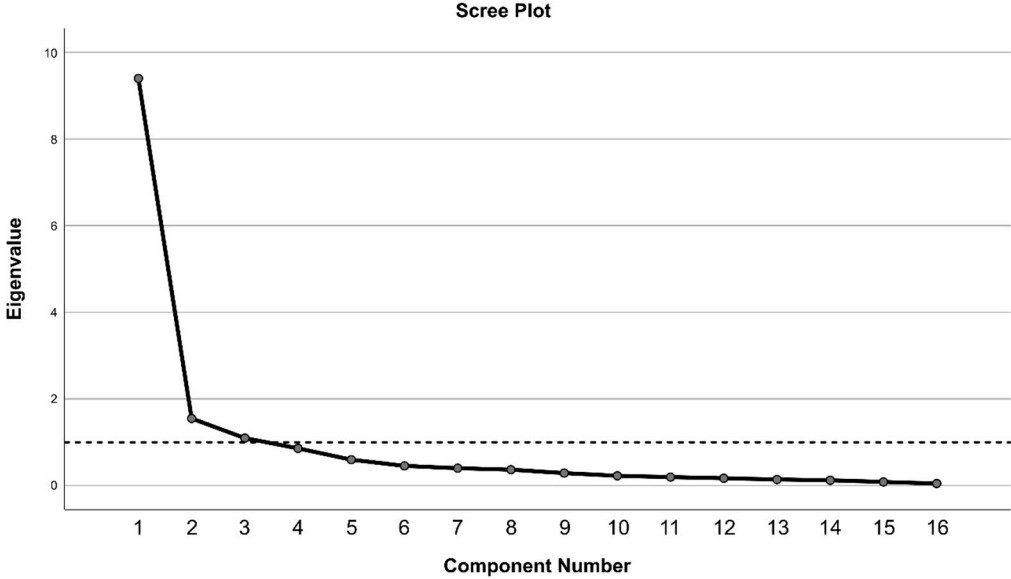

**Figure 1.** Scree plot of the eigenvalues in the factor analysis. The dashed line represents the reference eigenvalue (1).

Subsequently, for all domains, we employed a confirmatory factor analysis (CFA) to confirm factor loadings on each latent variable, and the relevant fit indices were expressed including comparative fit index (CFI), root mean square error of approximation (RMSEA), the Tucker–Lewis index (TLI), and chi square ($\chi^2$). The reliability of different domains was assessed using Cronbach's alpha ($\alpha$), composite reliability (CR), and average variance extracted (AVE). The independent associations among latent variables were tested to confirm or reject our hypothesis, and these were expressed as path coefficients. For the significant associations, we further tested impact of group-specific differences in parameter estimates based on the purpose of participants' visits to the destination. More specifically, groups were categorized as follows: 0 = work/education and 1 = other purposes, since the passengers from the former group might have stronger behavioral intentions to visit the

destination. A partial least squares multi-group analysis (PLS-MGA) was carried out in SmartPLS to test the difference in bootstrapping results as derived from each group [52].

**Table 1.** Pattern matrix of the factor analysis of the 16-item domain used to assess the perceived importance of safety measures at the airport.

| | Item | 1 | 2 | 3 |
|---|---|---|---|---|
| SL1 | Airlines have to sanitize airplanes after and before every flight | 0.957 | −0.16 | −0.073 |
| SL2 | Airport should locate a sanitizing gate or disinfection tunnel at all its entrances | 0.926 | −0.08 | −0.206 |
| SL3 | Airlines should leave at least 2 h between arrival and departure flights for sterilization and disinfection of aircrafts | 0.901 | 0.028 | 0.097 |
| SL4 | Airlines should close their sales' offices at airports | 0.866 | 0.087 | −0.005 |
| SL5 | Airport should put glass barriers to separate travelers from all service providers at the airport | 0.862 | −0.012 | 0.041 |
| SL6 | Airport should activate a safe route to transfer a traveler who is suspected to the quarantine zone | 0.86 | 0.076 | −0.024 |
| SL7 | Air conditioning systems at airport should be sanitized daily, and air purification technologies, such as plasma cluster technology, should be used. | 0.827 | 0.253 | −0.017 |
| ST1 | Airlines must give all travelers new face masks and gloves once they get into the airplane | −0.341 | 0.991 | 0.025 |
| ST2 | Travelers should wear face masks or face shields and gloves | −0.144 | 0.908 | 0.066 |
| ST3 | Airport should apply social distancing between the passengers in the waiting areas by leaving an empty seat between travelers | 0.122 | 0.891 | −0.139 |
| ST4 | All airport staff and crew of the airplane should have continuously renewed COVID-19 certificate | −0.144 | 0.876 | −0.014 |
| ST5 | All airport staff should be obligated to wear protective coveralls to protect them from infections | 0.126 | 0.875 | −0.237 |
| ST6 | All airport staff should wear face masks or face shields and gloves during work hours | 0.113 | 0.847 | −0.06 |
| IN1 | Airport should apply electronic payment applications for all services | −0.078 | −0.141 | 0.987 |
| IN2 | Airport should depend on robots instead of humans in some services at the airport | 0.033 | 0.004 | 0.822 |
| IN3 | Airport should carry out the disinfection and sterilization operations using environmentally friendly materials | −0.017 | 0.172 | 0.801 |

## 4. Results

### 4.1. Descriptive Statistics

A total of 954 passengers responded to the survey during the assigned study period. More than half of them were females (61.01%) aged 25–44 years (53.46%). The majority of the respondents had attained a high school degree or a bachelor's degree (71.70%, Table 2). Recreation for the pursuit of leisure activities was reported as the main purpose for visiting the destination by more than half of the respondents (53.2%), followed by visiting family members and relatives (25.2%), as well as cultural and educational purposes (17.6% and 15.7%, respectively, Figure 2).

**Table 2.** Demographic characteristics of the participants (*n* = 954).

| Parameter | Category | Frequency | Percentage |
|---|---|---|---|
| Gender | Male | 372 | 38.99 |
| | Female | 582 | 61.01 |
| Age | 18–24 | 182 | 19.08 |
| | 25–34 | 248 | 26.00 |
| | 35–44 | 262 | 27.46 |
| | 45–54 | 208 | 21.80 |
| | 55–64 | 42 | 4.40 |
| | >65 | 12 | 1.26 |
| Level of education | No formal education | 24 | 2.52 |
| | High school | 270 | 28.30 |
| | Bachelor's | 414 | 43.40 |
| | Masters | 54 | 5.66 |
| | Doctorate | 192 | 20.13 |
| The frequency of travels before the COVID-19 pandemic | I never travelled before | 90 | 9.43 |
| | Once | 288 | 30.19 |
| | 2–3 times | 336 | 35.22 |
| | 4–5 times | 90 | 9.43 |
| | >5 times | 150 | 15.72 |

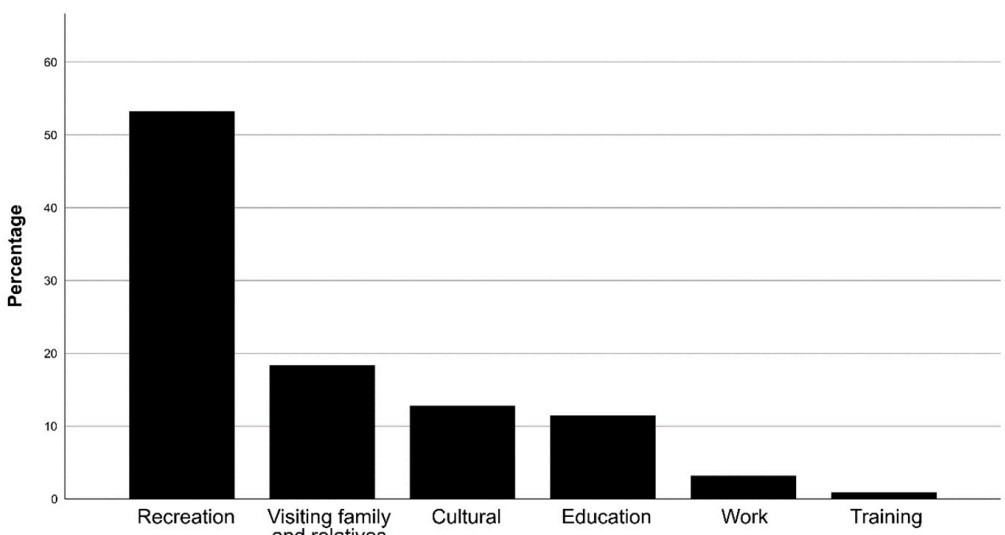

**Figure 2.** The percentages of participants' responses regarding their purpose of visiting the destination (*n* = 954).

### 4.2. Internal Consistency of the Study Instrument

The internal consistency of different subscales was tested using the Cronbach's alpha coefficient. The reliability of the questionnaire regarding the importance of safety measures was excellent (0.974). Furthermore, it was considered excellent for the domains of sanitization procedures and quarantine (0.967) and staff- and traveler-related preventive measures (0.969) and robust for innovative preventive measures (0.818) [53]. The reliability estimates for other domains of the survey are provided in the Table 3.

**Table 3.** Outer loadings and mean values of the indicators and the reliability of constructs included in the study instrument.

| Factors and Items | Mean (SD) | Standardized Factor Loading | α | AVE | CR |
|---|---|---|---|---|---|
| **Travel Risk Perception** | | | 0.974 | 0.726 | 0.920 |
| TRP1: Tourism can significantly increase the spread of SARS-CoV2 | 3.48 (1.10) | 0.795 | | | |
| TRP2: Tourism will be substantially influenced by the spread of SARS-CoV2 | 4.02 (0.41) | 0.889 | | | |
| TRP3: The risk of transmitting the infection increases by staying in a hotel | 2.97 (1.08) | 0.974 | | | |
| TRP4: Travel should be restricted to prevent the spread of the disease | 3.74 (1.20) | 0.775 | | | |
| TRP5: Currently, it is irresponsible to go on a business trip to highly endemic countries | 2.08 (0.97) | 0.879 | | | |
| TRP6: Currently, it is irresponsible to go on leisure trips to highly endemic countries | 3.57 (1.09) | 0.783 | | | |
| **Sanitization and logistics operations** | | | 0.976 | 0.779 | 0.962 |
| SL1: Airlines have to sanitize airplanes after and before every flight | 4.67 (0.79) | 0.841 | | | |
| SL2: Airport should locate a sanitizing gate or disinfection tunnel at all its entrances | 4.52 (0.85) | 0.897 | | | |
| SL3: Airlines should leave at least 2 h between arrival and departure flights for sterilization and disinfection of aircrafts | 4.67 (0.85) | 0.957 | | | |
| SL4: Airlines should close their sales' offices at airports | 4.45 (0.89) | 0.791 | | | |
| SL5: Airport should put glass barriers to separate travelers from all service providers at the airport | 4.33 (1.06) | 0.893 | | | |
| SL6: Airports should activate a safe route to transfer a traveler who is suspected to the quarantine zone | 4.67 (0.73) | 0.987 | | | |
| SL7: Air conditioning systems at airport should be sanitized daily, and air purification technologies, such as plasma cluster technology, should be used. | 4.00 (1.16) | 0.792 | | | |
| **Staff- and traveler-related measures** | | | 0.969 | 0.777 | 0.942 |
| ST1: Airlines must give all travelers new face masks and gloves once they get into the aero plane | 4.73 (0.77) | 0.897 | | | |
| ST2: Travelers should wear face masks or face shields and gloves | 4.66 (0.82) | 0.977 | | | |
| ST3: Airports should apply social distancing between the passengers in the waiting areas by leaving an empty seat between travelers | 4.32 (1.02) | 0.879 | | | |
| ST4: All airport staff and crew of the airplane should have continuously renewed COVID-19 certificate | 4.60 (0.85) | 0.794 | | | |
| ST5: All airport staff should be obligated to wear protective coveralls to protect them from infections | 4.36 (0.99) | 0.858 | | | |
| ST6: All airport staff should wear face masks or face shields and gloves during work hours | 4.77 (0.77) | 0.874 | | | |

| Factors and Items | Mean (SD) | Standardized Factor Loading | α | AVE | CR |
|---|---|---|---|---|---|
| **Innovative measures** | | | 0.854 | 0.730 | 0.855 |
| IN1: Airport should apply electronic payment applications for all services | 4.03 (1.13) | 0.793 | | | |
| IN2: Airport should depend on robots instead of humans in some services at the airport | 3.96 (1.01) | 0.846 | | | |
| IN3: Airport should carry out the disinfection and sterilization operations on a daily 24-h basis, using environmentally friendly materials and in accordance with the highest global safety and health standards | 3.40 (1.22) | 0.919 | | | |
| **Intention to travel** | | | 0.812 | 0.769 | 0.885 |
| INT1: I intend to revisit Sharm El Sheikh soon | 3.28 (0.82) | 0.789 | | | |
| INT2: I intend to revisit the city for work in the short/medium term, if needed. | 4.24 (0.76) | 0.927 | | | |
| INT3: I intend to revisit the city for leisure in the short/medium term | 2.12 (0.76) | 0.909 | | | |
| **Willingness to change or cancel travel plans** | | | 0.915 | 0.793 | 0.948 |
| WTT1: My travel behavior is likely to change due to coronavirus | 2.40 (0.60) | 0.937 | | | |
| WTT2: If I travel to another country depends on how media is reporting about that country | 2.20 (0.53) | 0.910 | | | |
| WTT3: Currently, I would cancel travel plans to countries with reported cases of coronavirus | 2.22 (0.60) | 0.967 | | | |
| WTT4: Currently, I would cancel travel plans to countries with no reported cases of coronavirus | 1.87 (0.54) | 0.814 | | | |
| WTT5: Currently I would avoid trips by airplane/boat | 2.15 (0.66) | 0.903 | | | |
| WTT6: Currently I would avoid trips by train | 2.14 (0.68) | 0.799 | | | |

*4.3. The Measurement Model (Confirmatory Factor Analysis)*

The reliability of individual items was used to validate the measurement model. As demonstrated in Figure 3, the outer loadings of all indicators to their respective latent constructs were above 0.707, as previously suggested [54], and they were all significant ($p < 0.05$). Furthermore, as shown in Table 4, the CR of different domains were acceptable, since it exceeded the cut-off value of 0.7 as suggested by Nunnally and Bernstein [55]. Additionally, the convergent validity of constructs fulfilled the threshold of AVE > 0.5 [56], which indicates that each domain could explain more than 50% of the variance in the relevant indicators. Finally, the discriminant validity requirement was satisfied, given that the square root of the shared variance between the items and their respective domain (indicated in bold) was greater than the relationship between different constructs in the matrix (Table 4). Therefore, we concluded that the used instrument was valid and reliable, and it measured distinct and identifiable items.

**Table 4.** Validity and reliability of the measurement model.

| Construct | CR | AVE | Fornell–Larcker Criterion | | | | | |
|---|---|---|---|---|---|---|---|---|
| | | | TRP | L | ST | IN | INT | WTT |
| Travel risk perception (TRP) | 0.726 | 0.920 | 0.874 | | | | | |
| Sanitization and logistics operations (L) | 0.779 | 0.962 | 0.031 | 0.784 | | | | |
| Staff- and traveler-related measures (ST) | 0.777 | 0.942 | −0.245 | 0.124 | 0.914 | | | |
| Innovative measures (IN) | 0.730 | 0.855 | −0.427 | 0.014 | 0.245 | 0.719 | | |
| Intention to travel (INT) | 0.769 | 0.885 | −0.124 | −0.475 | −0.512 | 0.171 | 0.841 | |
| Willingness to change or cancel travel plans (WTT) | 0.793 | 0.948 | 0.014 | −0.357 | −0.010 | −0.074 | 0.048 | 0.701 |

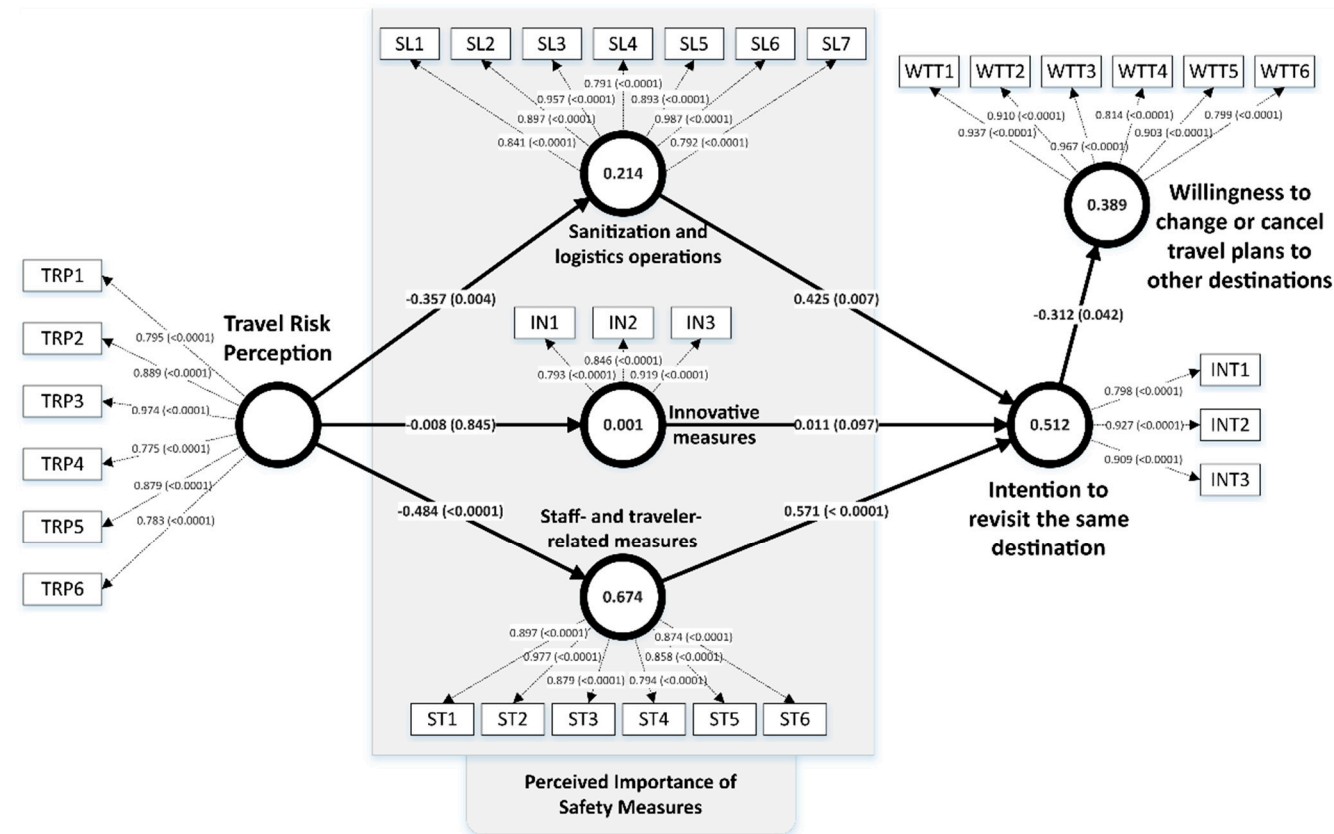

**Figure 3.** The percentages of participants' responses regarding their purpose of visiting the destination (*n* = 954).

### 4.4. The Structural Model

The relationship between the latent variables was estimated and validated in the structural model (Figure 3). The variance inflation factor (VIF) value was <5 for all analysis, indicating the lack of multicollinearity. The relationships between endogenous variables were significant ($p < 0.05$) except the relationship between travel risk perception and the perceived importance of innovative preventive measures and the intention to travel in the future and the importance of innovative measures.

Regarding $R^2$ values, the results revealed significant correlations between the constructs ($R^2 > 0.100$) as suggested by Falk and Miller [57], except the relationship between passengers' risk perception and innovative preventive measures ($R^2 = 0.001$). The correlation between risk perception and sanitization measures was judged as weak ($R^2 = 0.214$), since it did not reach the recommended cut-off value of 0.330 [58]. Furthermore, the com-

posite effect of the perceived importance of safety measures on respondents' intention to travel was moderate ($R^2$ = 0.512, Figure 3).

### 4.5. Results of the Primary Outcomes

The perceived risk of travel has negative effects on the perceived importance of sanitization and logistic operations performed by the airport to reduce the likelihood of COVID-19 spread ($\beta$ = −0.36, $p$ = 0.004) as well as the staff- and traveler-related preventive measures ($\beta$ = −0.48, $p$ < 0.0001). Additionally, respondents' intention to travel in the future was positively influenced by the perceived importance of sanitization procedures ($\beta$ = 0.42, $p$ = 0.007) and personal protective measures ($\beta$ = 0.57, $p$ < 0.0001). That is, if individuals perceive that such measures are very important, they exhibit greater intentions to revisit the destination. However, the implementation of innovative safety measures has no significant effect on the personal intentions to travel. Therefore, hypotheses 2a and 2c were supported by the current model, while hypothesis 2b was not accepted. Finally, the intention to travel to the destination in the future was inversely correlated with the willingness to cancel or change travel plans to other countries/places ($\beta$ = −0.312, $p$ = 0.04), which indicates that those who declared that they intented to revisit the destination would be less likely to cancel or change their future travel plans to other touristic places (Figure 3).

### 4.6. Between-Group Differences Based on the Purpose of Participants' Visits

The outcomes of the structural model were compared based on the purpose of passengers' purpose of visit, i.e., for work/education or other purposes (Table 5). The analysis was limited to the significant independent associations from the structural model. According to the Henseler's MGA approach [52], the positive independent association between the perceived importance of staff- and traveler-related safety measures and the intention to revisit the destination was significantly higher among passengers who had come for work/education as compared with those arriving for other purposes (the difference in the path coefficient = 0.17, $p$ = 0.041). No significant differences were detected between groups in other independent associations.

**Table 5.** Multi-group analysis by the purpose of visit.

| Relationship | Path Coefficients | | Path Coefficient Difference | $p$ * |
| --- | --- | --- | --- | --- |
| | Work/Education | Others | | |
| TRP-SL | −0.391 | −0.342 | −0.049 | 0.478 |
| TRP-ST | −0.497 | −0.477 | −0.020 | 0.874 |
| SL-INT | 0.441 | 0.397 | 0.044 | 0.140 |
| ST-INT | 0.648 | 0.478 | 0.170 | 0.041 |
| INT-WT | −0.357 | −0.301 | −0.056 | 0.371 |

\* $p$ value of Henseler's partial least squares multi-group analysis (PLS-MGA).

## 5. Discussion

The COVID-19 outbreak has dramatically influenced the global economy and the preferences of travel destination due to the uncertainty of exposure to risks and the differences in the geographic distribution of endemic regions. Perceived risk is an important construct, in which the potential dangers associated with a trip are associated with changes in the intention to revisit a given destination and/or changes in the behavioral control of traveling to other regions [59]. Although health risks have primarily affected tourists' behaviors and choice of destinations where an infectious disease is endemic, there is scant evidence regarding such intentions in the context of the widespread pandemic. In the current study, two constructs from the perceived importance of health safety measures at the Sharm El Sheikh airport have influenced passengers' future decisions to revisit the destination, which induced significant changes in the willingness to change or cancel travel plans to other touristic places.

The perceived risk of travel has negatively affected the importance of personal and logistic safety measures, with more robust effects on staff- and traveler-related safety precautions. The reported negative relationship in our model might have emerged from the lack of trust regarding the applied safety measures among individuals with higher levels of risk perceptions. This might be explained by the findings of Lee and co-authors, who revealed that tourists during the H1N1 pandemic might have exhibited adaptive behaviors to reduce the threat of infection to an acceptable level; thus, their adjusted behaviors might have reduced their levels of perceived risks and supported their travel decisions [46]. Recent studies of different populations in the Middle East have shown that good infection control measures, the existence of a reliable health system at the destination, and the use of hand sanitizers and face masks were the most significant factors that could impact travelers' decisions when choosing a trip [60–62]. A "new" characteristic of passengers' behaviors has also entailed hygienic precautions in the hotels rather than their service quality, location, or size [60]. Seemingly, during a pandemic, safety measures surpass other destination-specific attributes, such as operator performance, personal values, and consumer needs, as drivers of intention to re-travel or visit other destinations [63].

Therefore, the impact of safety procedures is a crucial variable in airport performance and passengers' satisfaction after re-opening the aviation. As indicated in our analysis, the importance of these measures is highly perceived by those coming for relatively more obliged purposes, such as work and education as compared with other purposes. Other demographic determinants of passengers' perceptions have been reported elsewhere, such as age, gender, and economic levels, which have changed the perceived importance. However, adopting health safety procedures in airports remains an important aspect for all categories of passengers.

Collectively, we stressed that a vital aspect in our model was to support efforts aimed at reviving the demand for tourism. Although the knowledge of passengers' perceptions as a whole has not been fully elucidated, our study contributes significantly to an invaluable domain of tourism recovery during the ongoing crisis. By providing insights into the factors that influence the intention to travel during the pandemic, decision makers in the tourism industry should be able to establish effective marketing strategies based on stressing the relevant safety procedures that are being implemented at the destination. This should, in turn, increase tourism demand and assist in the recovery of the national economies, which would also an integral part of the wider aspect of governmental actions to help mitigate the consequences of the worldwide pandemic. Therefore, the perceived risks of infection and barriers of travel should be targeted via optimizing quality seals and airport reputation, as well as communicating the measures of safety, hygiene, and cleanliness which are applied by relevant airlines.

Although post-pandemic research has shown significant concerns and uncertainty in the daily lives of consumers, people still have a more favorable attitude towards travelling, and only people with excessive anxiety and highly perceived risks would be less likely to travel [60,61,64]. Interestingly, since social media is an important driver of provoking fear due to misinformation, government and tourism decisionmakers are required to communicate with travelers via these platforms to convey the accurate situation regarding safety precautions in order to reduce the heightened perceptions of risks. This might include communicating trusted reports provided by independent experts to confirm the safety of airports and implementing promotional activities to reduce travelers' uncertainty.

Despite the inclusion of a large sample of air travel passengers, the present study may be limited by the survey-based design, which is subject to reporting bias and the inability to draw robust correlations. The study was also performed in a single airport in Egypt, which may limit the generalizability of the outcomes to other touristic regions. The study could be replicated in other countries to enable effective comparative analysis and to explore the perceptions of tourists from different cultures. The timeline of the study was only limited to the pre-vaccine period; therefore, the obtained results are subject to change due to altered perceptions when a vaccine becomes available. Finally, the convenience of

passengers regarding strict safety measures were not explored, which could be a matter of future research.

## 6. Conclusions

In conclusion, the applied operational and personal safety measures at the Sharm El Sheikh airport have played an important role in future intentions to revisit the destination as well as the perceptions of travelers towards traveling to other touristic places in the future. The inherently perceived risk of acquiring the COVID-19 infection was associated with low perceived importance/efficacy of safety measures and high inclination to future travel. The model applied in the present study was well-fitted and validated for future use, and it paves the way for similar studies in other regions. Airport safety should be acknowledged while implementing future strategic plans by governments, tourism stakeholders, and relevant authorities in order to control the perceived risks of tourists. Future investigations may include passengers of domestic and international flights. Furthermore, studies may consider the role of safety measures at multiple airports, hotels, and touristic sites and on airplanes to obtain more reliable and generalizable conclusions.

**Author Contributions:** Conceptualization, T.H.H.; Data curation, A.E.S.; Formal analysis, T.H.H. and A.E.S.; Funding acquisition, T.H.H.; Investigation, A.E.S.; Methodology, T.H.H. and A.E.S.; Project administration, A.E.S.; Resources, T.H.H. and A.E.S.; Software, T.H.H.; Supervision, T.H.H.; Validation, A.E.S. and T.H.H.; Visualization, T.H.H. and A.E.S.; Writing—original draft, T.H.H.; Writing—review and editing, T.H.H. Both authors have read and agreed to the published version of the manuscript.

**Funding:** The authors acknowledge the Deanship of Scientific Research at King Faisal University for the financial support under Nasher Track grant no. 206182.

**Data Availability Statement:** Data available on request due to privacy/ethical restrictions.

**Conflicts of Interest:** The authors declare no conflict of interest.

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
