# Peer review of "The Importance of Safety and Security Measures at Sharm El Sheikh Airport and Their Impact on Travel Decisions after Restarting Aviation during the COVID-19 Outbreak"

_sustainability, doi:10.3390/su13095216_

Round 1

Reviewer 1 Report

This paper presents a study of the impact of the safety measures at airports on travel decisions using the survey data at Sharm el Sheikh Airport. The paper is well written and the results can be useful in improving airport operations during the pandemic. My comments to further improve the paper are as follows:

  1. Based on my understanding, the authors mainly focused on the safety measures implemented at airports. However, it seems that the authors tried to find some correlations between the measures at airports and travelers’ intentions to revisit the destination. While I would agree that the two may have some correlations, many other factors such as safety measures on airplanes and in hotels may be more important in traveler’s destination choices. It would be helpful if the authors can make their focus clear.
  2. The authors seem to suggest the results can be helpful to tourism. However, only about half of the surveyed travelers are for leisure activities. I wonder whether the results can be representative of tourist’s preference in this case. 
  3. Why this study only surveyed international travelers at the airport?

Reviewer 2 Report

[IATA34] citation errors on line 153

Figure 2 is incorrectly named as Figure 1.

Very interesting conclusions. However, the research concerned to people who have decided to travel. It is important to find the people who have canceled their trip and ask them why.

I believe that there are so few supplementary materials that it could be included in the content of the article, because the reader has to open two documents in order to understand Figure 2. 

Reviewer 3 Report

Interesting and well elaborated paper. Title of the fig. 2 needs just to be corrected, as it refers to the structural model. Otherwise, congratulations!

Round 2

Reviewer 1 Report

I believe that the authors have improved their paper and adequately addressed my previous comments.